# Inclusion of 8% alfalfa silage advances feeding time and improves late-finishing feed efficiency in pigs

Yu Chen[1,2], Xiaogang Zhao[1,2], Xionggui Hu[1,2], Ji Zhu[1,2], Huibo Ren[1,2], Huali Li[1,2], Lihua Cao[1,2], Qingming Cui[1,2], Yuan Deng[1,2], Zhicai Li[1], Zhongshan Wei[1], Weimin Jiang[1,2], Yingying Liu[1,2]*, Yinglin Peng[1,2]*, Chen Chen[1,2]*

**1** Hunan Institute of Animal and Veterinary Science, Changsha, Hunan, China, **2** Yuelushan Laboratory, Changsha, Hunan, China

* 2004chch@hunaas.cn (CC); hunaulyy_2025@hunaas.cn (YL); 13907487646@hunaas.cn (YP)

## Abstract

Feed is the largest cost in pig production, driving interest in alternative ingredients that maintain performance at lower expense. Alfalfa, a nutrient-rich and widely available forage legume, shows promise as a partial feed substitute. However, it remains unclear how partial replacement with alfalfa affects growth dynamics, feed efficiency, and feeding behavior across fattening stages in pigs. This study investigated the effects of substitution with 8% alfalfa or alfalfa silage on growth performance and feeding behavior in growing-finishing pigs over a 60-day fattening period (60–130 kg). A total of 24 pigs were randomly assigned to three diet treatments: commercial feed, 8% alfalfa substitution, or 8% alfalfa silage substitution. Weight, feed intake, feed conversion ratio (FCR), and feeding hour were measured and analyzed using linear mixed-effects models and generalized additive mixed models. Results showed no significant effects of alfalfa on overall weight, feed intake or FCR. However, Pigs fed commercial feed showed superior FCR during mid-fattening (weeks 2.8–5.2, $P < 0.05$) but worse FCR than alfalfa-fed groups during late-finishing phase (weeks 6.6–8, $P < 0.05$). Moreover, alfalfa silage altered animal behavior by shifting feeding hour by roughly three hours earlier ($P < 0.05$). In summary, substitution with 8% alfalfa or alfalfa silage can be incorporated without compromising overall pig performance. While, integrating alfalfa in later fattening stages could further improve feed efficiency and reduce costs. This supports the use of phase-specific feeding strategies and adoption of such practices could enhance both economic and environmental sustainability in commercial pig farming.

**Data availability statement:** All data supporting the findings of this study are included in the article and its supplementary materials. The raw datasets are available from Zenodo: https://doi.org/10.5281/zenodo.15875443.

**Funding:** This work was supported by the Yuelushan Laboratory Breeding Program (YLS-2025-ZY01002, Yinglin Peng; YLS-2025-ZY04051, Chen Chen), the Funding of Hunan Province Swine Industry Technology System (HARS-05, Chen Chen), the Open Research Fund of Hunan Province Key Laboratory (2017TP1030, Chen Chen), and the Agricultural Science and Technology Innovation Fund Project of Hunan Province (2025CX101, Yingying Liu). The funders had no role in study design, data collection and analysis, decision to publish, or preparation of the manuscript.

**Competing interests:** The authors have declared that no competing interests exist.

## Introduction

Feed accounts for the largest proportion of production costs in modern pig farming, often exceeding 60% of total inputs [1]. With the rising prices of cereal grains and protein meals, there is a strong incentive to identify alternative feed resources that can reduce costs while maintaining performance and animal welfare. Alfalfa (*Medicago sativa*), a perennial legume, is well known for its high protein content, balanced amino acid profile, vitamins, minerals, and a range of bioactive phytochemicals including flavonoids and saponins [2]. Its abundant biomass and relative affordability make it a potentially valuable partial substitute in swine diets, particularly in regions with high alfalfa production.

Despite its nutritional value, the high structural fiber content of alfalfa may reduce its digestibility and energy availability in pigs, as their monogastric digestive systems are not well adapted to efficiently ferment fibrous substrates in the large intestine [3]. To address this limitation, microbial fermentation of alfalfa has been proposed to break down complex fiber structures, reduce anti-nutritional factors, and enhance the bioavailability of nutrients and bioactive compounds [4–7]. Alfalfa silage products have been shown to improve nutrient digestibility and positively modulate gut microbiota, potentially improving growth performance and gut health in pigs [3,8,9].

Additionally, feeding behavior is a critical component of swine management and is closely linked to nutrient absorption, gut health, and even the social dynamics of pigs [10,11]. Dietary fiber has been shown to affect these behavioral parameters by increasing chewing activity, slowing down ingestion rates, promoting satiety, and altering meal patterns throughout the day [12]. Diets high in fermented fiber can improve palatability and reducing the physical bulk of the feed, thereby enhancing feed intake by stimulating appetite and promoting the production of beneficial gut metabolites [13–15]. Therefore, partial substitution of commercial diets with alfalfa or alfalfa silage may have complex effects on pigs' feeding behavior that require systematic evaluation. It is important to understand these feeding behavior responses, as abnormal behavior patterns may impair production efficiency.

Despite growing interest in dietary fiber and fermentation strategies, no studies to date have systematically examined how alfalfa and its fermented form affect growth and feeding behavior over time during the fattening phase. This highlights a key gap, particularly at moderate inclusion levels (e.g., 8%), which may provide a practical balance between cost and nutrition in commercial pig production.

Therefore, this study was designed to evaluate whether substitution with 8% alfalfa or alfalfa silage can achieve cost-effective pig production without compromising growth performance in fattening stage. Additionally, the study investigated how such dietary strategies might influence diurnal feeding time patterns, to better understand their acceptability and welfare implications. We hypothesized that substitution with 8% alfalfa or alfalfa silage will not impair overall growth performance in fattening stage, but may have an impact on animal feeding behavior.

## Materials and methods

### Animal ethics

All animal procedures in this study were conducted in strict accordance with the ethical guidelines for animal experimentation, with full consideration of the 3Rs principles (Replacement, Reduction, and Refinement). The protocol was approved by the Animal Care and Use Committee of the Hunan Institute of Animal Husbandry and Veterinary Medicine (Approval No. HIAVS21-2509-01).

### Animals and experimental design

A total of 24 DLY pigs (Duroc × [Landrace × Yorkshire]) were used in this study. All pigs were approximately 4 months old with an average initial body weight of 63.24 ± 5.14 kg. Equal numbers of castrated males (n = 12) and females (n = 12) were included. Animals were housed in a temperature- and humidity-controlled facility, with each treatment group allocated to a separate pen (8 pigs per pen). Pens were identical in size (2.8 m × 6.6 m), equipped with full slatted floors, automatic drinkers, and electronic feeding stations (Runnong Technology Co., Ltd., Shenzhen, China). Environmental temperature was maintained between 20–24 °C, with 12-h light/dark cycles. All pigs had *ad libitum* access to feed and water throughout the 60-day experimental period.

The alfalfa used in this study was harvested at the early flowering stage and sun-dried to a moisture content of approximately 65%. The dried forage was chopped into ~2 cm segments with a forage cutter. For the alfalfa meal treatment, the chopped material was further ground into a fine powder (alfalfa meal) and incorporated directly into the diet. For the alfalfa silage treatment, the alfalfa meal was mixed with a commercial fermentation agent containing lactic acid bacteria powder and cellulase, sealed in airtight plastic bags, and ensiled for 60 days. The silage was subsequently dried at 60 °C and ground into a powder before being mixed with the diet. Feed mixing and processing for all diets were conducted using standard procedures at a commercial feed mill. The Chemical composition of alfalfa and alfalfa silage was shown in Table 1. The ingredient composition and proximate nutritional profiles of the three diets were provided in Table 2.

Pigs were randomly assigned to one of three dietary treatments (n = 8 per group), with each group comprising 4 castrated males and 4 females: Group A: commercial feed (control); Group B: commercial feed with 8% alfalfa meal substitution; Group C: commercial feed with 8% alfalfa silage substitution (Fig 1). Group B (alfalfa diet) and Group C (alfalfa silage diet) were designed by partially replacing corn, soybean meal, rice bran meal and wheat flour with 8% alfalfa meal or alfalfa silage meal, respectively, while maintaining similar levels of digestible energy and crude protein across treatments.

Daily individual-level measurements of body weight (kg), feed intake (kg), and feeding time were automatically recorded using electronic feeding stations. Feed intake relative to body weight was calculated daily by dividing feed intake (kg/day) by the corresponding body weight (kg). Biweekly feed conversion ratio (FCR) was calculated every 14 days as the total feed intake divided by total body weight gain over each interval. Outliers with FCR values below 1 or above 7 were excluded as biologically implausible. The experimental unit was the individual pig for all statistical analyses.

**Table 1. Chemical composition of alfalfa and alfalfa silage.**

| Feed | ME (MJ/kg) | DM (%) | CP (%) | EE (%) | NDF (%) | ADF (%) |
|---|---|---|---|---|---|---|
| Alfalfa | 14.51 | 96.41 | 14.77 | 0.36 | 36.87 | 27.55 |
| Alfalfa silage | 14.92 | 97.14 | 15.44 | 3.07 | 43.93 | 30.99 |

ME: Metabolizable energy, DM: dry matter, CP: crude protein, EE: ether extract, CA: crude ash, NDF: neutral detergent fiber, ADF: acid detergent fiber. Note: DM, CP, and EE contents were determined according to the procedures of the Association of Official Analytical Chemists (AOAC, 2005), using the oven-drying method (DM, 930.15), Kjeldahl method (CP, 954.01), Soxhlet extraction (EE, 920.39) respectively. NDF and ADF values were adopted from previous published study (Xu et al., 2023). ME of alfalfa and alfalfa silage was estimated using the predictive equation of Noblet and Perez (1993).

**Table 2. The ingredients and chemical composition of the commercial diet (feed A), alfalfa diet (feed B), and alfalfa silage diet (feed C) formulated for the experiment.**

| Ingredient | Feed A (%) | Feed B (%) | Feed C (%) |
|---|---|---|---|
| Corn | 60.00 | 57.00 | 57.00 |
| Soybean meal | 19.00 | 17.00 | 17.00 |
| Rice bran meal | 12.00 | 8.00 | 8.00 |
| Wheat flour | 6.00 | 7.00 | 7.00 |
| Alfalfa meal | 0.00 | 8.00 | 0.00 |
| Alfalfa silage meal | 0.00 | 0.00 | 8.00 |
| Dicalcium phosphate | 0.40 | 0.40 | 0.40 |
| Limestone powder | 0.30 | 0.30 | 0.30 |
| Salt | 0.30 | 0.30 | 0.30 |
| Vitamin-mineral premix (2%) [a] | 2.00 | 2.00 | 2.00 |
| Total | 100.00 | 100.00 | 100.00 |
| **Chemical composition** | **Feed A** | **Feed B** | **Feed C** |
| Digestible energy (DE, MJ/kg)[b] | 14.80 | 14.86 | 14.81 |
| Dry matter (DM, %)[c] | 95.95 | 97.64 | 96.44 |
| Crude protein (CP, %) [c] | 19.37 | 18.68 | 18.28 |
| Ether Extract (EE, %) [c] | 1.62 | 3.35 | 2.23 |
| Crude Fiber (CF, %) [c] | 2.63 | 3.79 | 3.29 |
| Crude Ash (CA, %)[c] | 8.08 | 6.78 | 6.26 |
| Calcium (Ca, %) [d] | 0.26 | 0.37 | 0.35 |
| Total phosphorus (TP, %) [d] | 0.44 | 0.42 | 0.42 |
| Available phosphorus (AP, %) [d] | 0.2 | 0.19 | 0.19 |

[a]Purchased from Hunan Lifeng Biotechnology Co., Ltd (Changsha, China). Supplied per kilogram of diet: 19.8 mg CuSO4.5 H2O; 0.20 mg KI; 400 mg FeSO4.7 H2O; 0.56 mg NaSeO3; 359 mg ZnSO4.7 H2O; 10.2 mg MnSO4·H2O; 5 mg vitamin K (menadione); 2 mg vitamin B1; 15 mg vitamin B2; 30 µg vitamin B12; 135 µg vitamin A; 2.75 µg vitamin D3; 0.45 µg vitamin E; 80 mg choline chloride.

[b]Digestible energy of the experimental diets was estimated using the predictive equation of Noblet and Perez (1993), based on the chemical composition (CP, EE, and fiber) of each diet.

[c]Values were chemically analyzed in laboratory. Dry matter (DM), crude protein (CP), ether extract (EE), crude fiber (CF), and ash (CA) contents were determined according to the procedures of the Association of Official Analytical Chemists (AOAC, 2005), using the oven-drying method (DM, 930.15), Kjeldahl method (CP, 954.01), Soxhlet extraction (EE, 920.39), Weende fiber method (CF, 978.10), and muffle furnace incineration (ash, 942.05), respectively.

[d]Calcium (Ca, %) of each diet was calculated as the sum of calcium contributed by each ingredient, weighted by its inclusion level, using standard values from NRC (2012). Total phosphorus (TP, %) was calculated as the sum of phosphorus in each ingredient weighted by its inclusion level, and available phosphorus (AP, %) was estimated by multiplying TP by ingredient-specific availability coefficients, based on NRC (2012) nutrient values for swine.

## FCR analysis

To assess the effects of different feed treatments on growth performance, we applied linear mixed-effects models (LMMs) using the lme4 package (Bates et al., 2015). For each response variable (body weight, feed intake, and feed intake relative to body weight), the following linear mixed-effects model was used:

$$Y_{ij} = \beta_0 + \beta_1 \cdot FeedType_i + u_i + \varepsilon_{ij}$$

where $Y_{ij}$ is the response variable (weight, feed intake, or feed intake/body weight) for pig $i$ on day $j$, $\beta_0$ is global intercept, $\beta_1$ is fixed effect of feed type (commercial, alfalfa, or alfalfa silage), $u_i \sim N(0, \sigma_u^2)$ is random intercept for individual pig $i$,

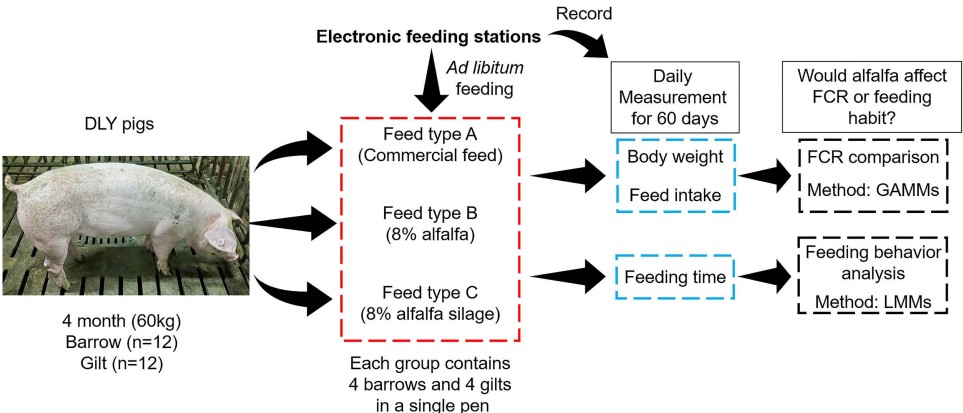

**Fig 1. Summary of study design.** All DLY pigs had ad libitum access to feed and water throughout the 60-day experimental period. Animals were randomly assigned to one of three dietary treatments (n = 8 per group), with each group comprising 4 barrows and 4 gilts: Group A – commercial feed; Group B – commercial feed with 8% alfalfa meal substitution; Group C – commercial feed with 8% alfalfa silage meal substitution. Daily measurement of weight, feed intake and feeding time was recorded by the station and further use for FCR analysis and feeding behavior analysis.

$\varepsilon_{ij} \sim N(0, \sigma^2)$ is the residual error. Model fit was evaluated using Type II Wald chi-square tests via the car package. The significance of fixed effects was determined using likelihood ratio tests, with $\chi^2$ statistics, degrees of freedom (df), and associated *P*-values reported.

To observe how FCR was changed over time, temporal trends for each group were analyzed using Generalized additive mixed models (GAMMs) via *mgcv* R package [16] and change rate were estimated via *gratia* R package to highlight stage-specific patterns over time.

The trait was modeled as:

$$Y_{ij} = \beta_0 + \beta_1 \cdot \text{FeedType}_i + f_j(\text{Day}_{ij}) + u_i + \varepsilon_{ij}$$

where $Y_{ij}$ is the response for pig *i* on day *j*, $\beta_0$ is the intercept, $\beta_1$ is the fixed effect of feed type (commercial, alfalfa, or alfalfa silage), $f_j(\cdot)$ is a smooth function of experimental day stratified by feed type (i.e., interaction smooths), $u_i \sim N(0, \sigma_u^2)$ is the random intercept for pig *i*, $\varepsilon_{ij} \sim N(0, \sigma^2)$ is the residual error.

## Feeding time analysis

Feeding time was analyzed both as a straight timeline (linear) and as a repeating 24-hour cycle (circular). The linear analysis allowed for the evaluation of overall feeding trends over time, capturing changes in feeding behavior along a continuous temporal scale. In contrast, the circular analysis addressed the periodic nature of feeding patterns, where the 24-hour cycle repeats, enabling the detection of daily rhythms and shifts in behavior that may not be apparent in a linear framework. Timestamp data were processed using the lubridate [17] and dplyr [18] packages and converted to decimal hours and radians.

To assess daily variation in feeding time, LMMs were used:

$$\text{Feeding Hour}_{ij} = \beta_0 + \beta_1 \cdot \text{FeedType}_i + u_i + \varepsilon_{ij}$$

where Feeding Hour$_{ij}$ is feeding time (hour of day) for pig$_i$ at observation$_j$, $\beta_0$ is global intercept, $\beta_1$ is fixed effect of feed type (categorical: A/B/C), $u_i \sim N(0, \sigma_u^2)$ is random intercept for individual pig *i*, $\varepsilon_{ij} \sim N(0, \sigma^2)$ is the residual error. Models were fitted using the lme4 [19] package and validated using residual analysis.

Circular statistics were conducted using the circular package [20]. Feeding hours were converted to angular measurements (0–2π radians). For each group, the mean feeding direction and concentration (ρ) were estimated.

## Statistical analysis

All statistical analyses were performed in R version 4.4.2 [21]. The individual pig served as the experimental unit for all analyses. Time-dependent traits were modeled using GAMMs from the mgcv package with random intercepts and feed-type-specific smooth terms. To account for temporal autocorrelation in repeated measures, four candidate correlation structures were tested within the GAMM framework: first-order autoregressive (AR1), compound symmetry, continuous-time autoregressive (CAR1), and autoregressive moving average (ARMA[1,1]). Based on Akaike Information Criterion (AIC) and likelihood ratio tests, the ARMA(1,1) structure yielded the best model fit and was selected for all final models. Model diagnostics were conducted using residual plots, autocorrelation (described above), and Q–Q plots. Group differences in smooth trends were assessed using: ANOVA to compare nested GAMMs (with vs. without interaction smooths); First derivatives of smooth functions to evaluate rates of change; Difference smooths to visualize localized treatment effects. For circular statistics, Rayleigh's test was used to assess the uniformity of the distribution, while Watson's two-sample test compared group means. A higher ρ indicated more clustered feeding behavior. All statistical significance was set at $P < 0.05$.

## Results

### Growth performance and feed intake across the fattening period

Over the 60-day fattening period, DLY pigs exhibited a general increase in body weight and daily feed intake but feed intake relative to body weight showed variable patterns over time (Fig 2). The final mass and weight gain for pigs with feed type A were 131.2 ± 10.8 kg and 66.4 ± 5.7 kg, respectively. Pigs on feed type B exhibited a final mass of 125.3 ± 11.5 kg and a weight gain of 63.9 ± 10.8 kg. Pigs receiving feed type C showed a final mass of 134.2 ± 7.3 kg and a weight gain of 72.0 ± 6.6 kg. LMMs revealed no statistically significant effects of feed type on most measured traits (Table 3). However, trends toward significance were observed for total feed intake ($\chi^2(2) = 5.02$, $p = 0.081$), feed intake relative to body weight ($\chi^2(2) = 5.14$, $p = 0.076$), suggesting possible biological effects of alfalfa on FCR that warrant further exploration (Table 3).

### FCR patterns and biweekly comparison

Analysis of FCR using GAMMs (Table 4) demonstrated no statistically significant differences in the overall average FCR trajectories between feed types. However, the smooth terms for biweekly periods were highly significant across all diets, indicating substantial biweekly fluctuations in FCR over time (Table 4). The GAMM model for FCR showed moderate explanatory power, with an adjusted $R^2$ of 0.577 based on 119 observations (Table 4).

Next, we analyzed FCR dynamics (Fig 3) with their change rate (Fig 4), and compared FCR patterns across feed types (Fig 5) to investigate localized differences. The FCR generally increased over time in pigs fed with feed A. In contrast, both alfalfa (Feed B) and alfalfa silage (Feed C) decreased FCR between weeks 4 and 6 in DLY pigs (Fig 3). The analysis of FCR change rate showed similar results. The change rate of FCR remained positive in commercial feed group (Feed A), while alfalfa (Feed B) and alfalfa silage (Feed C) group showed negative change rate from weeks 4 to 6 (Fig 4). Next, we compared FCR patterns across feed types to find out which time period were significantly different (Fig 5). The average FCR trajectories was significantly lower (P < 0.05) in commercial feed group (Feed A) compared to alfalfa (Feed B) and alfalfa silage (Feed C) groups from weeks 2.8 to 5.2 (Fig 5AB), while being significantly higher (P < 0.05) in commercial feed group (Feed A) than in alfalfa (Feed B) and alfalfa silage (Feed C) groups from weeks 6.6 to 8 (Fig 5AB). In addition, alfalfa (Feed B) showed significantly lower (P < 0.05) FCR than that in alfalfa silage (Feed C) group from weeks 1.0 to 2.4 (Fig 5C).

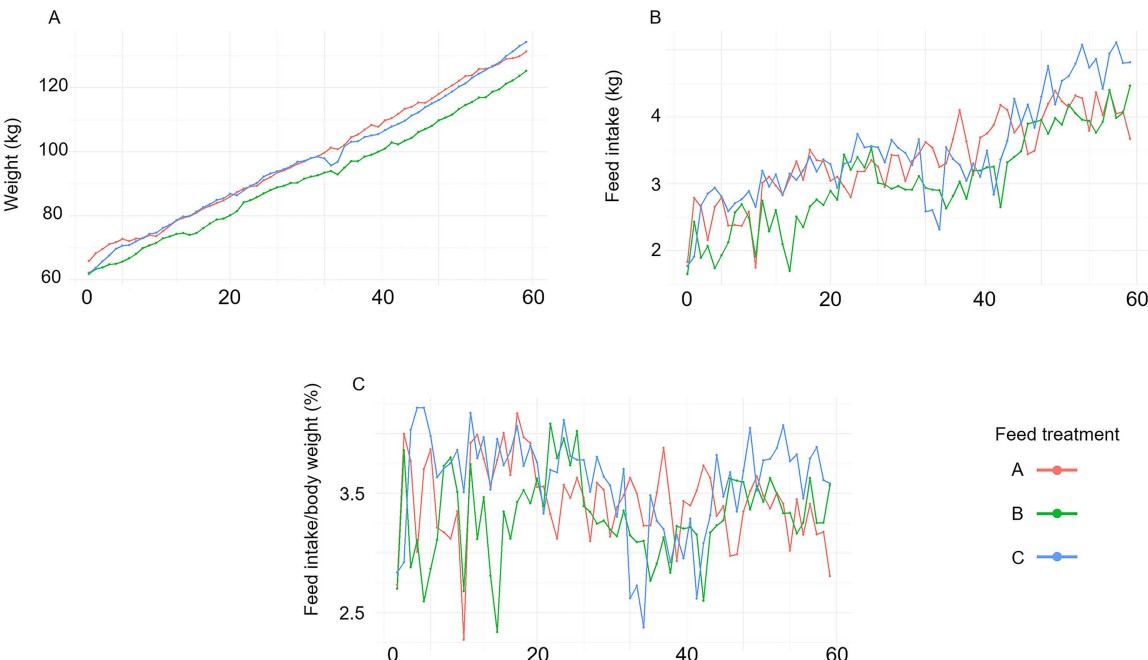

**Fig 2. Growth and feed intake trends over time.** During the 60-day fattening period, DLY pigs exhibited a general increase in body weight and daily feed intake. In contrast, feed intake relative to body weight displayed variable patterns over time.

**Table 3. Summary of fixed effects from linear mixed-effects models assessing the impact of feed treatment on pig weight, feed intake and feed intake/body weight.**

| Variable | Effect | $\chi^2$ | Df | P-value |
|---|---|---|---|---|
| Weight | (Intercept) | 1293.33 | 1 | P<0.001 |
| | Feed type | 3.18 | 2 | 0.203 |
| Feed intake | (Intercept) | 542.27 | 1 | P<0.001 |
| | Feed type | 5.02 | 2 | 0.081 |
| Feed intake/body weight | (Intercept) | 1459.02 | 1 | P<0.001 |
| | Feed type | 5.14 | 2 | 0.076 |

## Feeding behavior comparison

Finally, this study analyzed how alfalfa affect feeding behavior (feeding hour) in DLY pigs. The effects of feed treatments on pigs' feeding hour were evaluated using linear mixed-effects models and circular statistical methods (Table 5). The linear model estimated the average feeding time for pigs on the commercial feed (Feed A) to be around 1:22 PM. Pigs fed with alfalfa (Feed B) showed no significant difference in feeding time. However, pigs on alfalfa silage (Feed C) started feeding significantly earlier, around 10:15 AM ($p < 0.001$), with this shift becoming apparent within just five days (Fig 6).

Circular statistics supported these findings. Pigs on commercial feed (Feed A) had moderately consistent feeding times centered around 2 PM ($\rho = 0.422$), while pigs on alfalfa silage (Feed C) showed much more scattered feeding patterns ($\rho = 0.230$), indicating greater variability in when they ate. Pigs on alfalfa (Feed B) were slightly less consistent than those on commercial feed (Feed A) ($\rho = 0.366$) (Fig 7, Table 3).

**Table 4. Below table showed how FCR changed across time (every two weeks) for pigs on different diets, based on GAMMs.**

| Term | Estimate | Std. Error | t-value/ F | p-value | Interpretation |
|---|---|---|---|---|---|
| **Overall difference** | | | | | |
| Intercept (Feed A) | 3.673 | 0.137 | 26.89 | <0.001 | Baseline fcr |
| Feed B | −0.243 | 0.193 | −1.26 | 0.210 | No significant difference |
| Feed C | −0.172 | 0.193 | −0.90 | 0.373 | No significant difference |
| **Biweekly patterns** | | | | | |
| Feed A | – | – | 32.15 | <0.001 | Significant biweekly trend |
| Feed B | – | – | 18.86 | <0.001 | Significant biweekly trend |
| Feed C | – | – | 17.34 | <0.001 | Significant biweekly trend |
| **Model Fit** | | | | | |
| Adjusted R$^2$ | 0.577 | | | | Moderately high explanatory power |
| n | 119 | | | | Sample size |

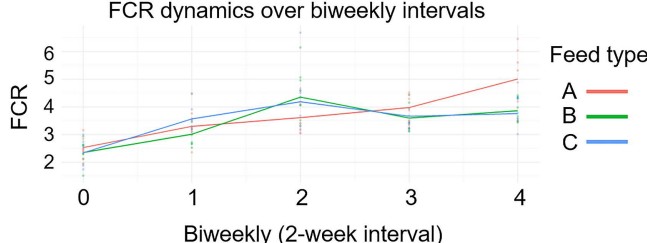

FCR dynamics over biweekly intervals

**Fig 3. FCR dynamics at biweekly intervals.** FCR generally increased over time in pigs fed commercial feed (Feed A). In contrast, pigs fed alfalfa (Feed B) or alfalfa silage (Feed C) exhibited a temporary decrease in FCR between weeks 4 and 6.

The Rayleigh test confirmed that feeding times were not random, with pigs tending to eat at specific times of day ($p < 0.001$). The Watson's test showed that the distribution of feeding times was significantly different across the three feed treatments ($p < 0.001$) (Table 4).

## Discussion

This study investigated the effects of replacing 8% of commercial feed with alfalfa or alfalfa silage on growth performance, feed intake, and feeding behavior over a 60-day fattening period (60–130 kg) in DLY pigs. Overall, the results indicated that alfalfa did not significantly affect overall performance, supporting the feasibility of partially substituting commercial feed with alfalfa-based products without compromising pig growth. However, FCR varied over time in DLY pigs fed alfalfa-based diets, with worse feed efficiency in the middle fattening phase and better efficiency toward the end. In addition, alfalfa silage feed advanced the pigs' feeding time by approximately three hours (Fig 8).

### Alfalfa inclusion alters feed efficiency dynamics during fattening

Consistent with previous studies, weight and feed intake are gradually increased over the fattening period, reflecting normal growth patterns in growing-finishing pigs [22,23]. Alfalfa had no significant impact on overall weight and feed intake, suggesting that pigs adapted well to 8% alfalfa and alfalfa silage feed. Previous work has indicated that pigs can tolerate moderate levels of dietary fiber without major adverse impacts on performance [24–26], which aligns with this study.

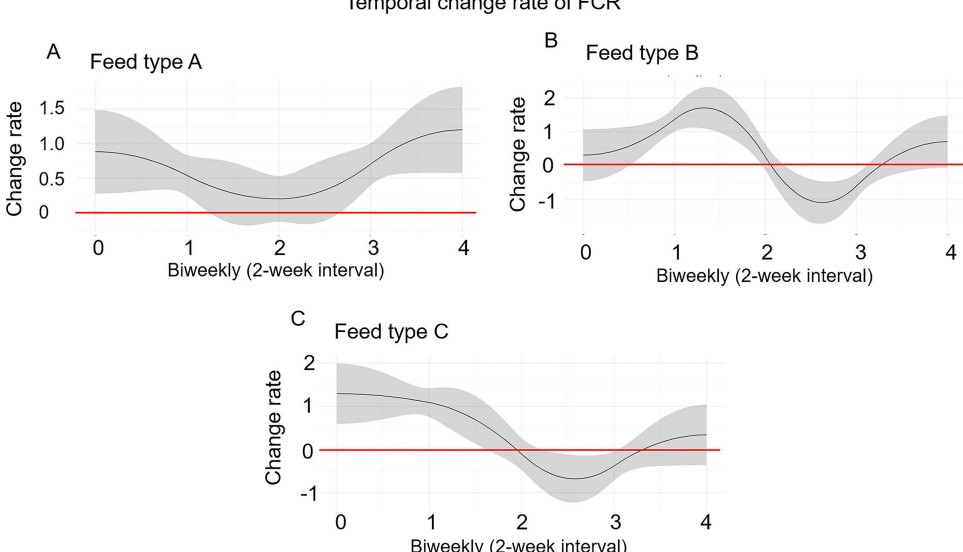

**Fig 4. Change rate of FCR by feed type.** The rate of FCR change remained positive in the commercial feed group (Feed A), whereas negative change rates were observed in both the alfalfa (Feed B) and alfalfa silage (Feed C) groups between weeks 4 and 6.

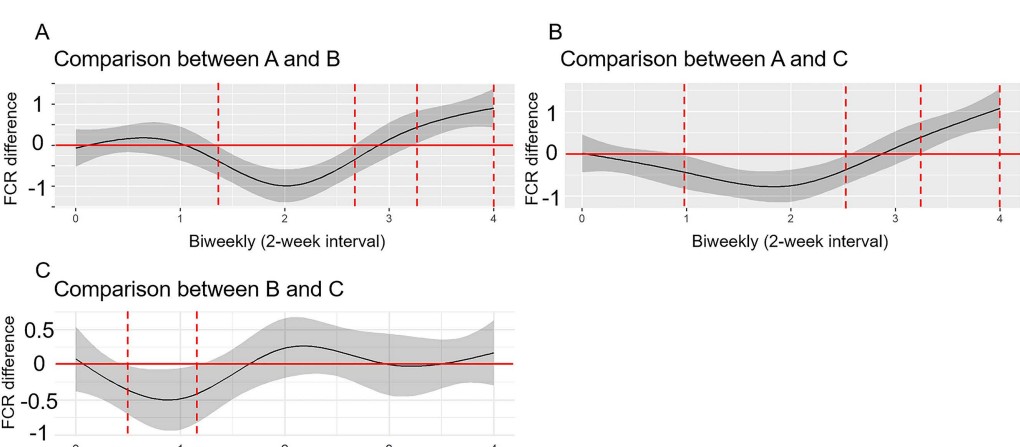

**Fig 5. Comparative FCR trajectories across feed types.** Average FCR was significantly lower (P < 0.05) in the commercial feed group (Feed A) than in the alfalfa (Feed B) and alfalfa silage (Feed C) groups between weeks 2.8 and 5.2. Conversely, Feed A exhibited significantly higher FCR (P < 0.05) than both alternative diets between weeks 6.6 and 8. Additionally, Feed B showed significantly lower FCR than Feed C between weeks 1.0 and 2.4.

While no significant overall differences in average FCR were found among alfalfa feed groups, the temporal dynamics of FCR varied substantially across the fattening period. Both alfalfa and alfalfa silage groups exhibited significant biweekly fluctuations in FCR, indicating that feed efficiency changed over time in a feed type–specific manner. For example, in the mid-fattening phase (weeks 2.8 to 5.2), pigs fed the commercial diet showed significantly lower FCRs compared to those receiving either alfalfa or alfalfa silage, suggesting better feed efficiency with commercial feed during this stage. This

**Table 5. Effects of feed treatment on pig feeding hour: linear mixed model and circular statistics results.**

| Model/ Measure | Estimate | Std. Error | t-value/ F | p-value | Interpretation |
|---|---|---|---|---|---|
| **Linear mixed model** | | | | | |
| Intercept | 13.37 | 0.18 | 73.1 | <0.001 | Mean feeding time ~13:22 (1:22 PM) |
| Feed B | −0.048 | 0.26 | −0.19 | 0.85 | No significant difference |
| Feed C | −3.22 | 0.26 | −12.45 | <0.001 | Significantly earlier feeding |
| Random SD | 0.485 | | | | |
| Residual SD | 5.808 | | | | |
| **Circular statistics** | | | | | |
| Feed A ($\rho$) | 0.422 | | | 0 | Moderate clustering, consistent around ~2 PM |
| Feed B ($\rho$) | 0.366 | | | 0 | Slightly less consistent |
| Feed C ($\rho$) | 0.230 | | | <0.001 | Highly variable, spread-out feeding times |
| Rayleigh test | | | 0.2097 | <0.001 | pigs tend to feed at specific times |
| Watson's test | | | 78.8 | <0.001 | Feeding times differ significantly by feed treatment |

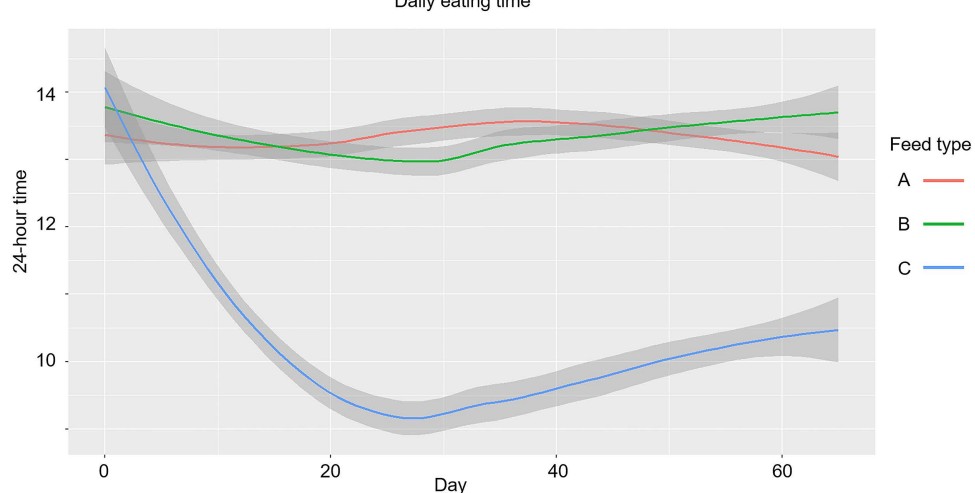

**Fig 6. A line chart showing feeding time shifts by diet.** The feeding time of pigs fed commercial feed (Feed A) and alfalfa feed (Feed B) remained around 1:22 PM. In contrast, pigs fed alfalfa silage (Feed C) began feeding earlier, with their feeding time shifting to approximately 10:15 AM within the first five days of the trial.

advantage may be attributed to the nutrient composition of the commercial diet, which is likely optimized for supporting rapid protein accretion and lean tissue growth characteristic of this period.

However, during the late fattening phase (weeks 6.6 to 8), this trend reversed. Pigs on the commercial feed exhibited significantly higher FCRs, while both alfalfa-based diets showed improved efficiency. This shift likely reflects the changing nutritional needs of pigs during the finishing stage, as energy requirements increase for fat deposition and maintenance rather than lean growth. The commercial formulation may be less suited to support this transition, while the fiber-rich alfalfa diets may improve nutrient absorption and satiety regulation [8] and modulate energy metabolism [8,27,28] under late-stage metabolic conditions. Additionally, a transient advantage of raw alfalfa over alfalfa silage was observed early in this study (weeks 1.0–2.4), where raw alfalfa resulted in significantly lower FCRs, potentially due to differences in fermentation byproducts or palatability influencing early intake and efficiency.

Circular visualization of daily feeding patterns

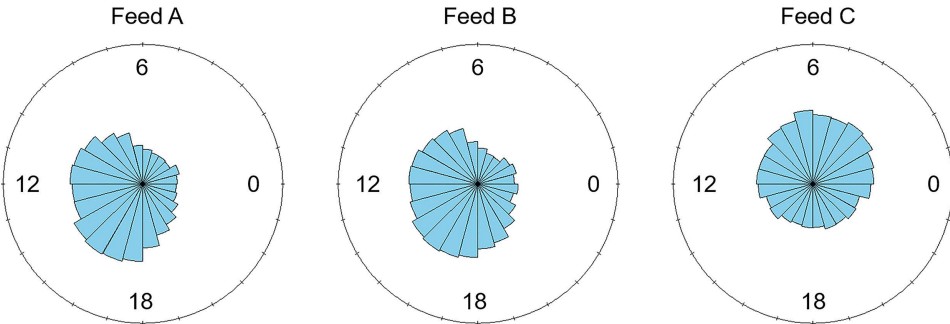

**Fig 7. Distribution of feeding hours by feed type.** Circular plots indicated that pigs fed commercial feed (Feed A) and alfalfa feed (Feed B) exhibited clustered feeding around 2 PM, while those fed alfalfa silage (Feed C) showed more dispersed feeding times, reflecting greater variability in feeding behavior.

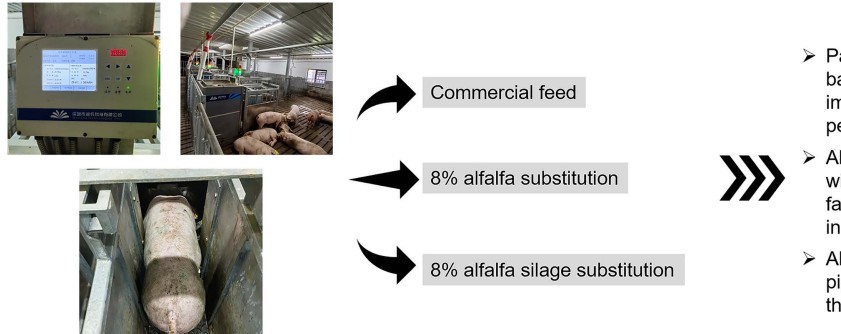

**Fig 8. Summary of dietary effects on pig performance.** Partial replacement of commercial feed with 8% alfalfa or alfalfa silage did not significantly alter most growth or feeding behavior parameters. Alfalfa-based diets resulted in poorer FCR during mid-fattening but improved FCR in later stages. Alfalfa silage feed advanced the timing of feed intake by approximately three hours, indicating behavioral changes without compromising overall performance.

These results support the rationale for phase feeding systems, where diet composition should be adjusted in accordance with the animal's developmental stage to maximize efficiency and carcass quality [29–31]. Given that feed costs account for 65–75% of total production expenses in swine operations, even marginal improvements in FCR can yield economically significant reductions in overall production expenditures [31].

Together, these findings highlight that phase-specific dietary effects should be considered when evaluating alternative feed ingredients. While commercial diets may optimize performance in early and mid-growth phases, incorporating alfalfa during late fattening can improve feed efficiency. These insights support the development of phase-feeding strategies to better align dietary formulation with pigs' changing physiological needs, potentially reducing feed costs and improving sustainability in commercial pork production.

## Alfalfa silage advances feeding time in pigs

Pigs fed alfalfa silage started eating about 3 hours earlier than others. This earlier feeding time shift could be beneficial in some production settings. Earlier feeding may align better with natural circadian rhythms of pigs, which are known to

exhibit morning-oriented feeding activity under housed conditions [32,33]. Feeding earlier in the day can help synchronize digestive and metabolic processes with daylight cycles, potentially improving nutrient utilization and promoting more stable energy metabolism throughout the active phase. In hot climates or during periods of heat stress, earlier feeding times may benefit pigs by shifting nutrient intake to cooler parts of the day, thereby reducing the risk of heat-induced declines in feed intake and growth performance [34,35]. Additionally, spreading feeding events earlier in the day may reduce feeder competition during peak hours, improving social stability and feeder access in group-housed pigs. In contrast, pigs fed with 8% alfalfa did not exhibit altered feeding time. This distinction reinforces the idea that feed processing method plays a critical role in shaping animal behavior.

However, it is worth noting that the higher variability in feeding hour (lower circular clustering) seen in pigs receiving alfalfa silage may suggest less predictable feeding patterns, which under restrictive feeding conditions could lead to social instability and inefficient feeder use [36,37]. Thus, while alfalfa silage may offer metabolic and behavioral advantages, future research should explore how to harness its benefits without compromising rhythmic consistency in feeding behavior.

The generalizability of the findings is constrained by the relatively small sample size (24 animals of a single breed) used in this study. Furthermore, the experiment was carried out under controlled conditions with ad libitum feeding, which may not accurately replicate typical field conditions. To enhance the applicability of the results, further research involving larger sample sizes or different breeds would be necessary. While the potential economic implications of incorporating alfalfa were acknowledged, a detailed economic analysis was not performed, as it fell outside the scope of the current investigation.

## Conclusions

Replacing 8% of commercial feed with alfalfa or alfalfa silage did not impair overall pig growth, supporting its use as a cost-effective, fiber-rich alternative. Feed efficiency varied by growth stage—poorer mid-phase FCR but improved late-phase efficiency with alfalfa—highlighting the value of using phase-specific diets. Alfalfa silage also shifted feeding time earlier, which may enhance nutrient use and reduce heat stress in certain systems. These findings suggest that incorporating alfalfa into precision feeding strategies may enhance both the sustainability and efficiency of pork production.

## Author contributions

**Conceptualization:** Yinglin Peng, Chen Chen.

**Data curation:** Weimin Jiang.

**Formal analysis:** Yu Chen, Xiaogang Zhao, Huali Li, Yingying Liu.

**Funding acquisition:** Yingying Liu, Yinglin Peng, Chen Chen.

**Investigation:** Yu Chen, Xiaogang Zhao, Xionggui Hu, Ji Zhu, Huibo Ren, Huali Li, Lihua Cao, Qingming Cui, Yuan Deng.

**Methodology:** Yu Chen, Xionggui Hu, Ji Zhu, Huibo Ren, Qingming Cui, Yuan Deng, Zhicai Li, Zhongshan Wei, Weimin Jiang, Yinglin Peng, Chen Chen.

**Resources:** Zhicai Li, Zhongshan Wei.

**Software:** Qingming Cui.

**Supervision:** Yingying Liu, Yinglin Peng, Chen Chen.

**Visualization:** Lihua Cao.

**Writing – original draft:** Yu Chen.

**Writing – review & editing:** Yu Chen, Xiaogang Zhao, Xionggui Hu, Ji Zhu, Huibo Ren, Huali Li, Lihua Cao, Qingming Cui, Yuan Deng, Zhicai Li, Zhongshan Wei, Weimin Jiang, Yingying Liu, Yinglin Peng, Chen Chen.

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
