## [Decision Letter · Decision Letter 0]

3 Dec 2025

Dear Dr. Chen,

Thank you for submitting your manuscript to PLOS ONE. After careful consideration, we feel that it has merit but does not fully meet PLOS ONE’s publication criteria as it currently stands. Therefore, we invite you to submit a revised version of the manuscript that addresses the points raised during the review process.

We look forward to receiving your revised manuscript.

Kind regards,

Andrey Nagdalian

Academic Editor

PLOS ONE

Journal Requirements:

Reviewers' comments:

Reviewer's Responses to Questions

**Comments to the Author**

1. Is the manuscript technically sound, and do the data support the conclusions?

Reviewer #1: Yes

Reviewer #2: Partly

2. Has the statistical analysis been performed appropriately and rigorously?

Reviewer #1: Yes

Reviewer #2: Yes

3. Have the authors made all data underlying the findings in their manuscript fully available?

Reviewer #1: Yes

Reviewer #2: Yes

4. Is the manuscript presented in an intelligible fashion and written in standard English?

Reviewer #1: Yes

Reviewer #2: Yes

Reviewer #1: All key results are correctly described. Perhaps, for completeness, it would be worthwhile to add explicit numerical totals of the final mass and total gain by group, so that the reader can directly assess the effect. However, the lack of such a proposal is not critical, since the figures and models overlap this information. The authors conscientiously interpret the absence of significant differences as the absence of a negative effect of alfalfa on productivity, and pay attention to the minor differences in time dynamics – this is a balanced approach without exaggerating the results.

The statistical analysis is beyond doubt. The only thing that can be recommended is to check the wording in the text again for typos related to statistics, for example, in line 1425, the phrase "time period were significantly different" should be corrected to "significantly different". Such small things do not affect the results, but it is important to eliminate them. In general, statistical data processing was carried out carefully and in full compliance with modern standards; the results reliably confirm the conclusions.

In the discussion section, it would be worth adding a couple of suggestions about the limits of the application of the results. For example, to emphasize the limited sample (24 animals of a single breed) and the fact that the results were obtained under controlled conditions with adlibitum feeding. This will help the reader understand that the conclusions are correct for these conditions, but may require testing on larger livestock or other breeds. It may also make sense to mention that the economic effect is implied, but was not directly calculated – such a remark will save from overloading conclusions with practical significance. Otherwise, the conclusions fully correspond to the data obtained and answer the questions posed in the introduction.

There are only minor typos and stylistic flaws in the text. The suffixes in adverbs were skipped several times – it should be "significantly different" instead of "significantly different" (line 1422) and "significantly lower" instead of "significantly lower" (line 1439). Also in line 1729, the word "In" is capitalized inside the sentence. These little things are easy to fix when editing. In general, the language and structure of the manuscript comply with the standards of scientific publications and, after minimal proofreading, will be understandable to a wide range of specialists.

The final recommendation is Minor revisions. The article is a technically high-quality study that meets the PLOS ONE criteria. Minor edits are needed to improve clarity – to clarify some details of the methods (feeding time), make minor language corrections, and discuss the limitations of the experiment. After these edits, the manuscript, in my opinion, will be ready for publication. There are no grounds for rejection or global recycling – the work fulfills its goals and provides valuable results for the pig feeding industry.

The manuscript deserves high praise for the novelty of the question, its rigorous experimental approach and the depth of statistical analysis. The score was lowered only due to a limited sample and some clarifications that need to be worked out. Otherwise, the research was carried out at a decent level.

Reviewer #2: The text should be carefully checked for typos and grammatical errors. The work is formatted using data indicating the relevance of the problem. The manuscript is technically sound, has sufficient sample sizes, controls and replicas that have undergone statistical processing. The data obtained may become outdated, since the relevance of this problem remains for a long time. It is advisable to refer to modern works touching on similar topics for the corresponding conclusions that are given.

**Do you want your identity to be public for this peer review?** For information about this choice, including consent withdrawal, please see our Privacy Policy

Reviewer #1: No

Reviewer #2: No

---

## [Author Response · Author response to Decision Letter 1]

5 Dec 2025

Reviewer #1: All key results are correctly described. Perhaps, for completeness, it would be worthwhile to add explicit numerical totals of the final mass and total gain by group, so that the reader can directly assess the effect. However, the lack of such a proposal is not critical, since the figures and models overlap this information. The authors conscientiously interpret the absence of significant differences as the absence of a negative effect of alfalfa on productivity, and pay attention to the minor differences in time dynamics – this is a balanced approach without exaggerating the results.

The statistical analysis is beyond doubt. The only thing that can be recommended is to check the wording in the text again for typos related to statistics, for example, in line 1425, the phrase "time period were significantly different" should be corrected to "significantly different". Such small things do not affect the results, but it is important to eliminate them. In general, statistical data processing was carried out carefully and in full compliance with modern standards; the results reliably confirm the conclusions.

In the discussion section, it would be worth adding a couple of suggestions about the limits of the application of the results. For example, to emphasize the limited sample (24 animals of a single breed) and the fact that the results were obtained under controlled conditions with adlibitum feeding. This will help the reader understand that the conclusions are correct for these conditions, but may require testing on larger livestock or other breeds. It may also make sense to mention that the economic effect is implied, but was not directly calculated – such a remark will save from overloading conclusions with practical significance. Otherwise, the conclusions fully correspond to the data obtained and answer the questions posed in the introduction.

There are only minor typos and stylistic flaws in the text. The suffixes in adverbs were skipped several times – it should be "significantly different" instead of "significantly different" (line 1422) and "significantly lower" instead of "significantly lower" (line 1439). Also in line 1729, the word "In" is capitalized inside the sentence. These little things are easy to fix when editing. In general, the language and structure of the manuscript comply with the standards of scientific publications and, after minimal proofreading, will be understandable to a wide range of specialists.

The final recommendation is Minor revisions. The article is a technically high-quality study that meets the PLOS ONE criteria. Minor edits are needed to improve clarity – to clarify some details of the methods (feeding time), make minor language corrections, and discuss the limitations of the experiment. After these edits, the manuscript, in my opinion, will be ready for publication. There are no grounds for rejection or global recycling – the work fulfills its goals and provides valuable results for the pig feeding industry.

The manuscript deserves high praise for the novelty of the question, its rigorous experimental approach and the depth of statistical analysis. The score was lowered only due to a limited sample and some clarifications that need to be worked out. Otherwise, the research was carried out at a decent level.

Response:

Thank you for your detailed and constructive comments. We are grateful for the positive feedback on the novelty, experimental approach, and depth of statistical analysis in our study. We have addressed all minor revisions as per the reviewer’s suggestions. We have made the necessary clarifications to improve the clarity of the methods, corrected the language issues, and incorporated the recommended discussion on limitations.

We have addressed the points as follows:

1. Typo Corrections:

Thank you for pointing out the typographical errors. We have carefully reviewed the entire manuscript and corrected the spelling as suggested.

2. Method section:

We have made revisions to the method section (feeding time) to enhance clarity, as follows: “The linear analysis allowed for the evaluation of overall feeding trends over time, capturing changes in feeding behavior along a continuous temporal scale. In contrast, the circular analysis addressed the periodic nature of feeding patterns, where the 24-hour cycle repeats, enabling the detection of daily rhythms and shifts in behavior that may not be apparent in a linear framework”.

3. Results section:

We acknowledge the reviewer’s suggestion to include explicit numerical totals of final mass and total gain by group. In response, we have added the detail in the result section as follow “The final mass and weight gain for pigs with feed type A were 131.2 ± 10.8 kg and 66.4 ± 5.7 kg, respectively. Pigs on feed type B exhibited a final mass of 125.3 ± 11.5 kg and a weight gain of 63.9 ± 10.8 kg. Pigs receiving feed type C showed a final mass of 134.2 ± 7.3 kg and a weight gain of 72.0 ± 6.6 kg”.

4. Discussion Section:

We appreciate the reviewer’s suggestion to discuss the limitations of the results in the context of the sample size and controlled conditions. In response, we have expanded the discussion as follows: “The generalizability of the findings is constrained by the relatively small sample size (24 animals of a single breed) used in this study. Furthermore, the experiment was carried out under controlled conditions with ad libitum feeding, which may not accurately replicate typical field conditions. To enhance the applicability of the results, further research involving larger sample sizes or different breeds would be necessary. While the potential economic implications of incorporating alfalfa were acknowledged, a detailed economic analysis was not performed, as it fell outside the scope of the current investigation.”

Reviewer #2: The text should be carefully checked for typos and grammatical errors. The work is formatted using data indicating the relevance of the problem. The manuscript is technically sound, has sufficient sample sizes, controls and replicas that have undergone statistical processing. The data obtained may become outdated, since the relevance of this problem remains for a long time. It is advisable to refer to modern works touching on similar topics for the corresponding conclusions that are given.

Response:

We greatly appreciate the constructive comments provided. In response, we have carefully reviewed the entire manuscript and made corrections to the typos and grammatical errors, as also suggested by Reviewer 1.

Regarding the suggestion to “refer to modern works touching on similar topics,” we would like to clarify that our discussion and reference list already include several recent and relevant studies on alfalfa, dietary fiber, and feeding behavior in pigs (e.g., Gao et al., 2024; Luo et al., 2021; Piles et al., 2025; Huangfu et al., 2024). These citations effectively connect our findings to contemporary research on the role of alfalfa in swine growth and precision feeding.

Our study specifically addresses a gap in the current literature by investigating the stage-specific effects of alfalfa substitution during the fattening period. While the long-term relevance of feed efficiency is acknowledged, our novel contribution lies in demonstrating that:

1. Overall performance is not compromised with 8% alfalfa inclusion.

2. Phase-dependent effects are observed (e.g., improved FCR in the late finishing phase for alfalfa groups), supporting modern precision feeding strategies.

3. Alfalfa silage significantly alters feeding behavior, particularly in terms of temporal patterns, with potential practical implications for farm management.

We believe that the manuscript already engages with the current scientific literature, and we hope this explanation effectively addresses the reviewer’s concerns.

---

## [Decision Letter · Decision Letter 1]

7 Jan 2026

Inclusion of 8% alfalfa silage advances feeding time and improves late-finishing feed efficiency in pigs

PONE-D-25-60915R1

Dear Dr. Chen,

We’re pleased to inform you that your manuscript has been judged scientifically suitable for publication and will be formally accepted for publication once it meets all outstanding technical requirements.

Kind regards,

Andrey Nagdalian

Academic Editor

PLOS One

Reviewers' comments:

Reviewer's Responses to Questions

**Comments to the Author**

Reviewer #1: All comments have been addressed

2. Is the manuscript technically sound, and do the data support the conclusions?

Reviewer #1: Yes

3. Has the statistical analysis been performed appropriately and rigorously?

Reviewer #1: Yes

4. Have the authors made all data underlying the findings in their manuscript fully available?

Reviewer #1: Yes

5. Is the manuscript presented in an intelligible fashion and written in standard English?

Reviewer #1: Yes

Reviewer #1: It is recommended to carefully proofread the text for individual stylistic details before the final publication, although in general the language of the article already meets high standards and does not contain the errors indicated by the reviewer. These remarks are advice for the final polishing of the text.

**Do you want your identity to be public for this peer review?** For information about this choice, including consent withdrawal, please see our Privacy Policy

Reviewer #1: No

---

## [Editor Report · Acceptance letter]

PONE-D-25-60915R1

PLOS One

Dear Dr. Chen,

I'm pleased to inform you that your manuscript has been deemed suitable for publication in PLOS One. Congratulations! Your manuscript is now being handed over to our production team.

Kind regards,

on behalf of

Dr. Andrey Nagdalian

Academic Editor

PLOS One